# The Nutritional Content of Rescued Food Conveyed by a Food Aid Organization

**DOI:** 10.3390/ijerph182212212

**Published:** 2021-11-20

**Authors:** Anne Nogueira, Fátima Alves, Paula Vaz-Fernandes

**Affiliations:** 1Centre for Functional Ecology-Science for People and the Planet, Department of Life Science, Faculty of Sciences and Technology, University of Coimbra, 3000-456 Coimbra, Portugal; fatimaa@uab.pt; 2Science and Technology Department, Universidade Aberta, 1269-001 Lisbon, Portugal; paulavaz@uab.pt; 3CAPP, Centre for Public Administration & Public Policies, Instituto Superior de Ciências Sociais e Políticas, Universidade de Lisboa, 1300-663 Lisbon, Portugal

**Keywords:** food insecurity, nutritional adequacy, food aid organization, rescued food

## Abstract

Background: The number of food-insecure families in the European Union has increased, resulting in an increasing number of households depending on food assistance programs. The aim in this study was to evaluate the nutrient content of food rescued by a food aid organization that rescues and redistributes fresh or freshly cooked food to low-income households. Methods: To determine the nutritional content of food hampers provided by our case study organization, we weighed all items of food hampers in three weighing rounds over a period of four months. The Food Insecurity Experience Scale (FIES) was applied to measure households’ food insecurity. Results: Our results show that, at our case study food aid organization, food donations substantially contribute to energy, macro, and micronutrient dietary recommendation intake (DRI). Conclusions: When evaluating how these nutrients contribute to alleviating food insecurity of the beneficiary households, we found that the perception of food insecurity is independent of the amount of nutrients served. To the best of our knowledge, this is the first study measuring the nutritional content of fresh or freshly cooked rescued food conveyed by a food aid organization.

## 1. Introduction

In 2017, the at risk of poverty rate in the European Union (EU-28) was 16.9% [1]. This means that the share of the European population with an equivalized disposable income below the at risk of poverty threshold (60% of the national median equivalized disposable income) was 16.9%. This means that of 119.1 million people, 42.5 million were not able to afford a quality meal every second day [2]; meanwhile, the amount of European food waste is still around 88 million tons per year [3]. 

Low-income individuals are often viewed as being linked to poverty and social exclusion since insufficient resources may generate a reduced ability to purchase adequate quantities of healthy nutritious food [4], resulting in situations of food insecurity [5].

Food security is said to exist “when all people, at all times, have physical, social, and economic access to sufficient, safe, and nutritious food that meet their dietary needs and food preferences for an active and healthy life” [6]. According to the Food and Agriculture Organization (FAO) (2009), the four pillars of food security are: (1) the availability of sufficient quantities of food of appropriate quality; (2) access by individuals to adequate resources for acquiring appropriate foods for a nutritious diet; (3) use of food through adequate diet, clean water, sanitation, and health care to reach a state of nutritional well-being; (4) stable access to adequate food at all times [7]. The nutritional dimension is integral to the concept of food security [6]. 

To assess food insecurity at the household level, a psychometric scale for assessing food insecurity the Food Insecurity Experience Scale (FIES)) can be used. The FIES has been used to identify the following risk factors of food insecurity around the globe: low levels of education, weak social networks, less social capital, low household income, and being unemployed [8,9]. In 2020, the FIES was applied to investigate gender implications in food insecurity, showing that in every European area, women are more fragile in terms of food insecurity than men [10]. Furthermore, the same study suggested that the number of children in the household have a stronger impact on women’s food insecurity than on men’s, and the factor that most mitigates food insecurity for both men and women is higher education [10].

As in other countries, the FIES was translated and validated into Portuguese [11]. In Portugal, it was applied by the General Directorate of Health in Portugal to monitor the food security of Portuguese households during the period of 2011–2014 [12]. The results showed that socio-economically vulnerable Portuguese populations appear not only to have diminished access to foods of high nutritional value but also, in some cases, show poor management of economic resources and family food. Women were found to be more susceptible to food insecurity than men, while elderly people were found to apparently be less food insecure, which might be explained by many of these elderly individuals having lived in a context of socio-economic precarity, and thus have developed adaptation and resilience mechanisms that lead to a lower perception of food insecurity. Furthermore, compensatory mechanisms to deal with isolation strongly influence the food consumption patterns of unemployed individuals and elderly people living alone. Gregório et al. (2017) also observed strong social cohesion in Portuguese society, which may be due to dense solidarity networks provided by social security, religious organizations, or civil society initiative [12]. 

Several initiatives worldwide tackle the issue of food poverty of low-income individuals through different governmental and nongovernmental organizations, such as food banks, soup kitchens, and food recovery and redistribution organizations that donate food or sell it for a symbolic price [13,14]. To alleviate both food poverty and food waste, multiple hunger relief organizations connect people in need with food waste or surpluses [15,16]. In this case, surpluses, or products close to the best-by date, usually originating from the food industry sector, the distribution sector, or the food service industry, hotels, restaurants, and cafes (HORECA) channel, are rescued and can be redistributed as-is or used to prepare meals at soup kitchens. Therefore, the nature and quantity of the food served by organizations that rely exclusively on donated food are not controlled by the organizations, as they depend on the nature and quantity of food donors´ surpluses each day [17]. The organizations face the daily challenge of both managing food sources with an everchanging nature and improving the nutrient intake of the recipients of the redistributed food [18,19,20,21].

As a consequence, the nutritional quality of emergency food has been examined by researchers in different charitable organizations in several high-income countries such as Canada [22,23,24,25,26], France [27], Germany [28], Italy [29], the Netherlands [30], the U.K. [31,32,33], the USA [34,35,36,37,38,39,40,41,42], and Australia [18]. In addition to the homeless, members of food-insecure households resort to food banks in emergencies [23,43,44] or use them as long-term food sources [45]. The longer low-income households depend on food assistance programs the more nutritional adequacy evaluations are crucial to determine if the programs meet the clients’ (long-term) needs [43]. 

During the past decades, we have been assisting with the proliferation of charitable meal programs as well as with a shift in the assisted population [26], both of which may have been created due to the 2008 worldwide economic recession at first, and then were more recently exacerbated by the COVID-19 pandemic, which has led to increasing numbers of previously well-off families all over the world experiencing scarcity of resources. As such, many of them had no other choice than to feed themselves with the help of food banks, food pantries, soup kitchens, urban farms, and other charitable organizations [46,47,48,49]. The COVID-19 pandemic is directly affecting food systems through impacts on food supply and demand, as lockdowns have prevented small-scale food producers from supplying their products to consumers, and indirectly through decreases in purchasing power [50]. 

The results obtained from research on European countries concur with those obtained in research in the North American continent [35,39]. In these studies where the nutritional intake of beneficiaries of food aid services was evaluated, the amount of macro and/or micronutrients was considered inadequate for the following reasons: (1) the lack of availability of the amounts and types of food to ensure a balanced diet; (2) poor food choices; (3) beneficiaries not knowing how to cook certain foods. 

Considering the previously mentioned at risk of poverty rate and the amount of wasted food in the EU and the EU policies to achieve the Sustainable Development Goals established by the United Nations [51], which include helping to eradicate hunger and malnutrition, food systems transformation, and saving nutritious food for redistribution to those in need, the research output on the nutritional value of upcycled food in the European continent has been unexpectedly low. This is precisely what we aimed to address in this study: (1) to quantify the nutrients of rescued and redistributed foods through a case study that involved the operations center of the Refood project in Leiria, Portugal [52]; (2) to evaluate how these nutrients contribute to alleviating the food insecurity of the beneficiary households. The Refood project is an independent, 100% volunteer-run organization that works to eliminate both food waste and hunger at the micro-local level [52], rescuing food waste to support low-income households who receive food hampers 1–3 times per week.

The novelty of this study concerns the nature of the rescued and redistributed food. Unlike other previously examined emergency food projects that rely mainly on dry staple foods or processed surpluses, most Refood-rescued and -redistributed food is freshly cooked (in restaurants, cafes, bakeries, and hotels) or is naturally fresh (fruit and vegetables).

## 2. Materials and Methods

A cross-sectional study was conducted from November 2018 to March 2019 during three rounds of five consecutive working days in intervals of two months for food data collection, and over 8 consecutive working days of May 2019 for sociodemographic data collection and FIES application, in 59 adults using a food rescue and redistribution organization. The study case organization, Refood-Leiria, was visited 23 times within the mentioned months. During the data collection process, all items served to all families were weighed. After this process, the 27 families comprising 59 individuals (47 adults and 12 children) covered by all three weighing rounds were selected to ensure a significant amount of data per family. Thus, a total of about 3500 items were weighed, representing a total mass of 1,721,642 kg; tares were deducted, and edible percentages of foods were calculated whenever relevant (e.g., fruit, vegetables, certain meats, and fish) (Appendix A—Total Methodology Flow Chart).

The collected food data were analyzed and converted into nutrients using Food Processor SQL, a computer program developed from the 1980s by ESHA researchers (Salem, OR, USA), with nutritional information from tables of food composition of the United States Department of Agriculture (USDA), and successively improved by new generations of researchers in several countries, to increase the quantity and nature of food involved [53]. In Portugal, this task has been performed since the beginning of the 1990s by researchers from the Department of Clinical Epidemiology, Predictive Medicine and Public Health of the Faculty of Medicine of the University of Porto and the Institute of Public Health to contain more and more culinary dishes that are typically Portuguese. 

Whenever the food distributed did not correspond to one of the foods listed in the Food Processor SQL manual, the dishes were composed from existing data (e.g., 180 g of typical migas from the western region of Portugal was broken down into 60 g cornbread, 60 g of cabbage, 60 g of black-eyed peas, and one tablespoon of olive oil).

The collected data were then systematized and coded to be introduced in the Food Processor, which calculates the macronutrients (proteins, lipids, carbohydrates, and dietary fiber), micronutrients (vitamins and minerals), and caloric intake. The results were then exported to IBM SPSS^®^ Statistics 27 (IBM Corporation, Armonk, NY, USA) to be statistically analyzed. 

The following nutritional parameters were analyzed: total energy intake, proteins, total carbohydrates, free sugars (sugars added to food and drink), total fat, saturated fatty acids, monounsaturated fatty acids, total polyunsaturated fatty acids, trans fatty acids cholesterol, polyunsaturated fatty acids omega 3 and omega 6, dietary fiber, vitamin A, thiamine, riboflavin, niacin, pantothenic acid, vitamin B_6_, folate, vitamin B_12_, vitamin C, vitamin D, vitamin E, vitamin K, calcium (Ca), copper (Cu), iodine (I), iron (Fe), magnesium (Mg), phosphor (P), potassium (K), sodium (Na), and zinc (Zn). The percentage dietary reference intake (DRI) of each studied nutrient provided by recovered food in our case study was calculated according to the following references: (1) for energy, macronutrients, dietary fiber, and micronutrients, we used values published by the Institute of Medicine [54]; (2) for cholesterol, free sugars, saturated fat, monosaturated fat, polyunsaturated fat, trans fatty acids, we used the WHO recommendations [55]; (3) for fiber, the American Heart Association recommended values were used [56]. 

After the process of weighing of the food items contained in the hampers, we preliminarily evaluated each hamper’s content with respect to the relative mass percentage of each food group and no visible mass percentage differences were found [57]. Furthermore, the study case organization rescues and redistributes food surpluses from distribution, the HORECA channel, bakeries, and canteens, which means that from day-to-day, the nature and amount of the collected and redistributed food vary widely. The beneficiaries collect their bags, containing at least one meal (but very often more than one) plus snacks 1, 2, or 3 times per week according to their socio-economic situation or according to their availability to afford the trip to the operations center. When families have a very low income, the trip cost to the operation center may be a significantly high proportion of the family budget, so it is agreed with the family that the collection takes place fewer times each week, but the amount of redistributed food is as large as possible. Therefore, the sizes and contents of the collected bags are heterogeneous, even if the bag content is as balanced as possible [57]. Moreover, the same hamper may contain more than one kind of soup, main course, side dish, dessert, fruit, or snack, according to the daily assortment of rescued food. As a result, the percent DRI was calculated globally, considering the mean value of each nutrient provided per person and per collection day.

An appropriate instrument, the Food Insecurity Experience Scale (FIES), was developed in the 1990s by the United States Department of Agriculture (USDA),and has been translated to around 200 languages [58,59] and applied in several studies to measure food (in)security at the household level [60,61]. In a global context, FIES was applied to a sample of 150 countries with results positively correlating with FAO data concerning the prevalence of undernourishment, national income, health, and well-being [62]. The FIES scale was applied to the member of the beneficiary household who usually collects food at the Refood-Leiria operations center, and who, complementarily, also responded to a survey for the sociodemographic characterization of their household. The objective of the study was explained to all beneficiaries as well as that their participation was voluntary and that there would be no retaliation for refusing to participate in the survey. Food insecurity using the FIES psychometric scale was categorized according to the scale validation report [11].

Body mass index (BMI) was calculated according to reported values of weight and height. In the case of children, BMI was calculated using height and weight values reported by the parent, the sex and age of the child, together with the Centers for Disease Control and Prevention BMI Calculator for Children and Teen online tool [63].

Statistical analysis was performed using IBM SPSS^®^ Statistics 27. Descriptive statistics were used to summarize participant´s characteristics and to examine the level of food insecurity in the study sample. 

## 3. Results

Table 1 summarizes the sociodemographic sample. The 27 families in the sample consisted of 59 individuals (34 female and 25 male recipients), with 47 adults with a mean age of 47.7 ± 15.6 years old (seven individuals were over 65 years old), and 12 children with a mean age of 9.6 ± 4.3 years old. Households consisted of one to four members, of whom one to three were adults, and none to two were children. 

Among the adult members of the sample, there were 16 active members, 17 unemployed, one was a student, and 13 were retired. All 12 children attended school at the appropriate year level. The academic qualifications of the adults varied from two school years to a bachelor’s degree, with a mean value of 8.0 ± 3.4 school years. In terms of nationalities, there were 50 Portuguese and nine Brazilian respondents. 

The food insecurity categories results (Table 1), which were determined differently for households with and without children, were merged to evidence the totality of sample families experiencing each situation: food security (*n* = 3), low insecurity (*n* = 13), moderate insecurity (*n* = 3), and severe insecurity (*n* = 8). To classify and quantify the foods distributed by Refood in relation to the Portuguese food guidelines a 2200 Kcal reference diet was selected [64] and one major assumption was made concerning the definition of s a child. The UN Convention on the rights of children (1989) defined a child as “every human being below the age of eighteen years, unless under the law applicable to the child, majority is attained earlier”. The WHO defines adolescents as individuals in the 10–19 years age group [65]. However, for this study, we considered the volume of ingested food for a child over ten years old as similar to the food volume an adult ingests, remembering that the nutritional needs of a child/teenager are different from those of an adult [66,67]. 

The daily mean macro- and micronutrients of redistributed food and the percentage of DRI [54,55,56] for beneficiaries are presented in Table 2. The results show that the food hampers contributed to more than 100% of the DRI of Fe (156.3%), Cu (140.3%), total fat (133.7%), free sugar (128.5%), P (115.8%), folate (112.2%), and Zn (111.6%). 

Figure 1 shows that although the redistributed food did not meet the DRIs for the remaining nutrients, the food benefits still provided 91.1% of carbohydrates, 89.5% of dietary fiber, 86.1% of thiamine, 85.3% of niacin, 83.0% of protein, 79.8% of vitamin A, 76.7% of riboflavin, 76.2% of saturated fat, 74.8% of vitamin C, 72.8% of monounsaturated fat, 65.0% of Na, 64.6.0% of polyunsaturated fat, 64.4% of vitamin B_6_, 62.5% of vitamin B_12_, 60.9% of omega 6, 59.1% of vitamin K, 58.5% of cholesterol, 56.6% of energy, and 54.2% of Mg. Furthermore, the food donations supplemented recipients’ dietary intake with pantothenic acid (48.0%), omega 3 (41.7%), K (40.4%), Ca (29.9%), vitamin E (18.4%), I (7.3%), trans fat (2.9%), and vitamin D (1.5%). 

The reported values of weight and height were used to calculate the BMI of all beneficiaries. The mean adult BMI value was 25.9 ± 4.62, with 1 individual underweight, 22 having a healthy weight, 15 overweight, and nine obese. For children, BMI was calculated using height and weight values reported by parent, and the sex and age of the child, together with the Centers for Disease Control and Prevention BMI Calculator for Children and Teen online tool [63], which also provides the BMI categorization considering the child’s age and sex. No child fell in the underweight category, six had a healthy weight, two were overweight, and four were obese. For children, because, according to the child´s age and sex, the same BMI value may fall into different categories, no mean BMI value was calculated. 

This study was conducted focusing on families and not on individuals. Food (in)security was found to be directly related to the mean amount of energy contained in each family hamper. However, two of the four food (in)security category frequencies had less than five families; therefore, the four categories of food (in)security (food security, low insecurity, moderate insecurity, and severe insecurity) were regrouped into two (food security and low insecurity, and moderate and severe insecurity) to homogenize the distribution. Thus, the distribution of macro and micronutrient mean values into the two new food insecurity categories was found to not have a normal distribution. As a result, two non-parametric tests, the median test and the Mann–Whitney U-test, were applied (Table 3), showing no statistical differences (*p* < 0.05), except for polyunsaturated fat and omega 6.

## 4. Discussion

The nutritional results showed that rescued and redistributed fresh or freshly cooked food can contribute to low-income households’ DRI of energy, and macro- and micronutrients. As we were unable to find parallel studies that examined DRI of exclusively rescued and redistributed fresh or freshly cooked food, we could not extrapolate beyond this case study. Moreover, the results in the previously mentioned literature are often expressed in other units other the dietary reference intake (DRI). Still, we used the results from other researchers who studied food provided to low-income households by emergency food organizations for discussion purposes, regardless of their nature or national nutritional guidelines.

The estimate of energy provided by the food donation in the present research is 1244.4 Kcal, which is 56.6% of the DRI, higher than that reported Sparke et al. [31,32,37] but lower that those found by other researchers [37], which, in some cases, exceeded the recommendations [23,33,43]. As food hampers offered by Refood-Leiria are supposed to provide for at least one full meal, plus snacks, this caloric intake is adequate with respect to the Portuguese food guidelines of 2200 Kcal for the reference diet selected [64]. 

The mean supply of macronutrients in this case study not only almost met the DRI for protein and carbohydrates but exceeded the DRI with respect to total fat (Table 1). This concurs with the organization providing for at least one meal per day but, whenever possible, more than one meal is served to improve beneficiaries´ nutrition on days when they do not collect food from the organization. Other researchers found food-aid organizations supplying larger portions (>%DRI) of protein [33,37,43,68], carbohydrates [33,37], and total fat [32,39]. Still, some studies found insufficient amounts (<%DRI) of protein [23,25,36,41], carbohydrates [41], and fat [23,41] being served by emergency food programs. 

In addition to total fat, other types of conveyed fat were examined. The findings showed that the provision of each fat type was lower than the DRI. For saturated fat (DRI < 10% energy), trans-fat (DRI < 1% energy), and cholesterol (<300 mg, the equivalent of 58.8% DRI), all three results were adequate. Cholesterol-adequate food was also documented in a soup kitchen in the USA [39]. Similar results of lower-than-recommended polyunsaturated fat [41] and monounsaturated fat [37] have been found by other researchers. Conversely, saturated fat was found to be significantly higher that recommended in food aid organizations in the U.K. [31,33]. Omega 3 (41.7% of the DRI) and omega 6 (60.9% of the DRI) provisions were considerable, unlike reported in a study in the USA [37], which reflects the amount fish present in food rescued by the organization.

Our findings showed that the percentages of free sugar were higher than the DRI. The same result was found by researchers who examined the nutritional content of foods distributed by the Supplemental Nutrition Assistance Program in the USA [37,38], and by a food bank and a charitable meal service in the U.K. [31,33]. Even though nearly no sugary drinks are rescued and redistributed by our case study organization, they rescue a large amount of bakery products. These bakery products often contain free sugar as a part of the recipe or even as a leavening aid; therefore, a considerable quantity of these products is diverted to other organizations to avoid sugar-unbalanced dietary patterns in beneficiaries. Nevertheless, bakery products are served to be consumed as breakfast and mid-morning or mid-afternoon snacks in amounts appropriate for more than one day. 

Figure 1 shows that the dietary fiber provided by the food donation was 89.5% of the DRI, an adequate provision. Even if adequate provisions of fiber were found in two studies [33,68], most researchers reported inadequate amounts of served fiber [24,31,37,39,41]. One of the reasons that may explain the large amount of fiber served by the Refood-Leiria organization is the Portuguese tradition of vegetable soup consumption at nearly every meal, as well as of vegetable side dishes (usually salads or boiled vegetables). Hence, it is not surprising that these products are constantly present in food rescued from different food sources (the HORECA channel, the distribution sector, and canteens) and are thus served by the organization.

The redistributed mineral micronutrients depicted in Figure 1 have DRIs for Fe, Cu, P and Zn, all above 100%, mirroring the redistribution of meat and green-leaved vegetables. Adequate [24,39,68] and high [43] provision of Fe, and insufficient Zn [23,24,29,31,32] have been documented. The Na and Mg DRI provisions were >50% but lower than 70%, consistent with the small amounts of salt in cooked rescued food. The Na results are consistent with those from research of two food aid organizations [29,39], but Mg was found to be lower than in other organizations [23,24,31,32,41]. The pattern of low amounts fresh fruit and vegetable donations in many organizations is consistent with findings of the inability to meet K nutritional requirements [27,31,32,39,41]. However, the Refood-Leiria organization conveys large amounts and a wide variety of fresh fruit and vegetables, which, due to their nature, met 40.4% of the total K DRI, consistent with usually redistributed items such as jacket potatoes and bananas. Because few dairy products are rescued and redistributed by the Refood-Leiria organization, it is not surprising that the DRI for calcium is lower than 50%. However, this is evidence of how vegetables such as kale, a typical Portuguese soup ingredient or side dish, are good sources of calcium. The scarcity of donations of dairy products, fruit, and vegetables, causing the lack of Ca, has been reported in American and European research quantifying the nutritional content of food provided by charitable services [23,24,27,29,32,39,41,42,43,68]. The provision of I was found to be much lower than the DRI, unlike results documented in British food banks [33]. However, our results concur with the European iodine deficiency identified by the World Health Organization [69].

Food hampers distributed by our case study organization always contained fresh fruit and vegetables. As such, it is not surprising that the percent DRI of the calculated vitamin content was high, unlike most reported findings. Rescued and redistributed foods’ folate provision exceeded 100% DRI, a result previously documented in the USA [43,68]. Vitamin A, thiamine, riboflavin, niacin, and vitamin C were provided in amounts ranging from 75% to 100% of the DRI. Vitamin B_6_, vitamin B_12_, and vitamin K percent DRI provisions were lower than but still over 50%. The previously mentioned researchers observed a low or very low percent DRI of vitamin A [23,24,31,32,33,39,41,42,68], thiamine [24,27], riboflavin [23,24,27], niacin [23,27], vitamin C [23,32,41,42,68], vitamin B_12_ [23], and vitamin K [41], consistent with the identified insufficient provision of fruit and vegetables. Our findings of a DRI lower than 50% for pantothenic acid and vitamin E are consistent with the small amounts of rescued and redistributed seeds and nuts. Finally, vitamin D, seldom present in food, was expectedly found in very small amounts, and thus also in other food aid organizations [23,29,33,41]. Vitamin D, needed for human metabolism, is mainly synthesized by the human body when in contact with solar radiation [70,71].

The results of nutrient distribution between the two food insecurity categories were non-normal, with high standard deviation values, even after efforts to normalize the distribution through the application of conversion factors such as dividing each hamper’s contents by the number of individuals per household and the number of weekly collection days. A second strategy to normalize the distribution involved removing outlier households from the sample. None of the attempts to normalize the distribution were successful due to the everchanging nature and quantity of daily rescued food requiring the organization to redistribute the majority of what is collected within 24 h. Thereby, adequate nonparametric tests, the Mann–Whitney U-test and median test, were applied. The results identified no statistical difference for nutrients provision between the two food insecurity categories, except for two nutrients. The Mann–Whitney U-test, with a significance of 5%, showed significant differences only for polyunsaturated fat (*p* = 0.030) and omega 6 (*p* = 0.023), suggesting that the beneficiaries´ perception of food insecurity is independent of the amount of nutrients provided by the organization. This finding also suggested that the source of food insecurity might be related to other socio-economic factors, similar to other researchers´ findings [8,9]. Moreover, no significant differences were found when correlating households´ occupational status—steady income (active and retired) and unsteady income (unemployed and part-time workers)—with the two food (in)security categories with an exact Fisher’s test (*p* = 0.391). 

This research’s limitations may have been due to the non-normal nature of the distribution of the data, preventing the establishment of causal associations between food insecurity and the provided macro- and micronutrients and socioeconomic outcomes. One different limitation is related to assessing food insecurity based on the individual self-perception of the respondent, and the tool used does not allow the identification of how food insecurity affects each member of the household. Another f limitation is that no measure of food intake was possible, and thus there was no possibility of evaluating the contribution of donated food to the total diet. Finally, the reported values of height and weight are another acknowledged limitation.

Several strengths should also be noted related to the data collection process, which involved weighing of more than 3500 items. This study design captured the mean amount of 33 nutrients and energy, allowing the calculation of each nutrient’s percentage of DRI provided by rescued and redistributed food to low-income families. Furthermore, this was a pilot study on the contribution of upcycled freshly cooked food considering larger issues concerning food security, not only in Portugal, but from an international point of view. One key issue is that upcycled food, especially prepared food, is likely to be culturally appropriate and more likely eaten, providing quantities of food of appropriate quality for a nutritious diet. Secondly, the barriers to food security presented by a lack of cooking time, tools, or knowledge are avoided to some degree, allowing the use of food in all circumstances. Finally, our results challenge prior work showing low nutrition quality in many traditionally distributed shelf-stable or bulk food foods, which may contribute to changing the direction of research to more sustainable food security paths.

For future studies, a recommended design would include having a control group of families prior to being supported by the organization as well as conducting the research in other Refood operation centers, or in other similar organizations. The case study organization, Refood-Leiria, is one operation center of the national Refood organization that comprises 51 operation centers spread from North to South Portugal, one in Spain (Madrid), one in Italy (Milan), and one developing in the USA (Richmond, Virginia). All operation centers conduct similar activities at the micro-local level, consisting of rescuing fresh or freshy prepared food, mainly from the HORECA channel and the redistribution sector, to redistribute it to low-income families. Thus, despite the sample size, the generalization of the results may be possible in the Portuguese context. To confirm the national generalization, the next round of research may focus on a Refood operation center in another context such as a larger Portuguese city (Lisbon or Porto). Another line of research should focus on Refood’s activity in international contexts.

Another method of improving the research design would be to evaluate the beneficiaries’ total food intake, so that the contribution of donated food to the total diet could be calculated.

## 5. Conclusions

During the past decades, the food insecurity situation of the Portuguese population has been tackled by the public welfare system, but more recently, other governmental and some nongovernmental organizations such as the Refood project have immerged to support low-income households.

To the best of our knowledge, this is the first study to measure the nutritional content of fresh or freshly cooked rescued food by a food aid organization. Whereas other researchers have found that supplemental foods offered from food pantries are an important resource, a lack of sources of macro- and micronutrients provided by nonindustrial meals and fresh fruit and vegetables has been frequently documented. Conversely, our results showed that, for our case study, food aid organization, food donations substantially contributed to energy, and macro- and micronutrient DRI intake. When evaluating how these nutrients contribute to alleviating the food insecurity of the beneficiary households, we found that the perception of food insecurity was independent of the amount of nutrients served one to three times per week to each vulnerable household. Increasing the weekly frequency of hamper distribution requires food sources, physical space, refrigerating equipment, and human resources.

Food aid organizations require local cross-sector collaboration to achieve increased participation in food surplus donations from the HORECA channel and the distribution sector, to encourage volunteerism from university students, or to enact corporate social responsibility including volunteerism in continuing professional development.

The COVID-19 pandemic served as an example of a crisis that has posed unprecedented challenges to food loss and waste in the global food supply system as well as an increasing food insecurity risk to the population. We learned that, instead of resuming life as normal, life should transition to a new, more sustainable kind of normality where food recovery and redistribution to low-income families is a widespread practice of urban mining, fostering positive outcomes for biodiversity, human well-being, and climate.

Data from this study provide valuable information to increase the awareness of the nutritional value of food diverted from landfills. The findings also provide a strong argument to encourage policymakers not only to increase the resources allocated to organizations that both rescue food surpluses and fight food insecurity but also to implement policies that increase access to stable and sufficient quantities of adequate food to the food insecure.

## Figures and Tables

**Figure 1 ijerph-18-12212-f001:**
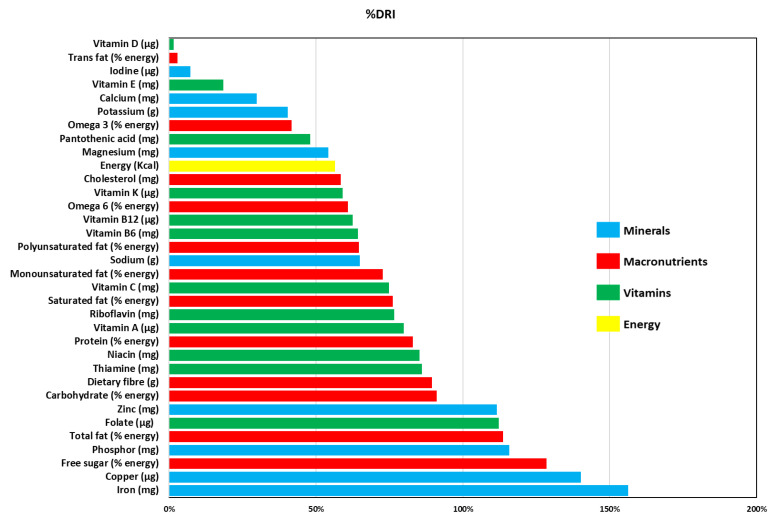
The percentages of the daily contribution of the food donations to the dietary reference intake (DRI) of Refood-Leiria beneficiaries. Institute of Medicine dietary reference intakes for energy, macronutrients, fiber, and micronutrients [54]; World Health Organization (WHO) dietary reference intakes for cholesterol, free sugars, saturated fat, monosaturated fat, polyunsaturated fat, trans fatty acids [55]; American Heart Association dietary reference intake for fiber [56].

**Table 1 ijerph-18-12212-t001:** Sociodemographic and anthropometric sample characterization (*n* = 59).

Sociodemographic Sample Characteristics		
	* **n** *	**%**
Families	27	
Households		
1 2 3 4	9 8 6 4	33.4 29.6 22.2 14.8
Households with children		
Yes No	8 19	29.6 70.4
Gender		
Female Male	34 25	57.6 42.4
Occupational status		
Active members Unemployed Students Retired	16 17 13 13	27.2 28.8 22.0 22.0
Highest Level of Education (adults, *n* = 47)		
1–4 years 5–6 years 7–9 years 10–12 years >12 years	12 7 12 15 1	25.5 15.0 25.5 31.9 2.1
Nationality		
Portuguese Brazilian	50 9	84.7 15.3
Food insecurity		
Food security Low insecurity Moderate insecurity Severe insecurity	3 13 3 8	11.1 48.2 11.1 29.6
Body mass index		
Underweight Healthy weight Overweight Obese	1 28 17 13	1.7 47.5 28.8 22.0

**Table 2 ijerph-18-12212-t002:** Mean value of nutrients distributed per person and per day and percentage of the dietary reference intake (DRI).

Mean Value of Nutrients Distributed per Person and per Day		
**Nutrients**	**Mean Value**	**% DRI**
Energy (Kcal)	1244.4	56.6
Protein (% energy)	18.68	83.0
Carbohydrate (% energy)	50.08	91.1
Free sugar (% energy)	13.16	128.5
Total fat (% energy)	31.27	113.7
Saturated fat (% energy)	9.16	76.2
Monounsaturated fat (% energy)	12.74	72.8
Polyunsaturated fat (% energy)	5.17	64.6
Trans fat (% energy)	0.04	2.9
Cholesterol (mg)	175.50	58.5
Omega 3 (% energy)	0.38	41.7
Omega 6 (% energy)	4.57	60.9
Dietary fiber (g)	24.6	98.4
Vitamin A (µg)	638.7	79.8
Thiamine (mg)	0.99	86.1
Riboflavin (mg)	0.92	76.7
Niacin (mg)	12.8	85.3
Pantothenic acid (mg)	2.4	48.0
Vitamin B6 (mg)	1.03	64.4
Folate (µg)	448.9	112.2
Vitamin B12 (µg)	1.5	62.5
Vitamin C (mg)	61.7	74.8
Vitamin D (µg)	0.23	1.5
Vitamin E (mg)	2.76	18.4
Vitamin K (µg)	62.0	59.0
Calcium (mg)	328.4	29.9
Copper (µg)	1262.5	140.3
Iodine (µg)	10.9	7.3
Iron (mg)	12.5	156.3
Magnesium (mg)	200.7	54.2
Phosphor (mg)	810.7	115.8
Potassium (g)	1.9	40.4
Sodium (g)	1.3	65.0
Zinc (mg)	10.6	111.6

**Table 3 ijerph-18-12212-t003:** Mean value and standard deviation of nutrients distributed per person, per day and per two food insecurity categories, and results of median test and Mann–Whitney U-test per nutrient.

Nutrients	Food Security and Low Insecurity	Moderate and Severe Insecurity	Mann–Whitney U-Test	Median Test
Energy (Kcal)	1540.7 ± 1810.5	813.6 ± 344.0	0.394	1.000
Protein (% energy)	18.44 ± 2.679	18.84 ± 4.755	0.394	0.440
Carbohydrates (% energy)	52.43 ± 8.291	46.84 ± 7.756	0.481	0.704
Free sugar (% energy)	12.97 ± 2.105	13.43 ± 2.819	0.394	0.704
Total fat (% energy)	29.35 ± 6.689	34.03 ± 4.184	0.099	0.252
Saturated fat (% energy)	8.564 ± 2.184	9.978 ± 1.554	0.148	0.252
Monounsaturated fat (% energy)	11.96 ± 3.171	13.87 ± 1.811	0.195	0.704
Polyunsaturated fat (% energy)	4.807 ± 0.9055	5.708 ± 1.075	0.030	0.252
Trans fat (% energy)	0.0556 ± 0.1305	0.1000 ± 0.0205	0.342	0.183
Cholesterol (mg)	187.3 ± 152.5	158.4 ± 103.0	0.865	0.704
Omega 3 (% energy)	0.3590 ± 0.162	0.3987 ± 0.1294	0.544	0.704
Omega 6 (% energy)	4.252 ± 0.859	5.032 ± 1.015	0.023	0.252
Dietary fiber (g)	34.81 ± 78.76	9.642 ± 9.998	0.089	0.120
Vitamin A (µg)	707.4 ± 537.0	538.7 ± 312.7	1.000	1.000
Thiamine (mg)	1.26 ± 1.29	0.614 ± 0.249	0.110	0.440
Riboflavin (mg)	1.087 ± 0.853	0.666 ± 0.284	0.318	0.440
Niacin (mg)	14.64 ± 9.802	10.25 ± 7.318	0.342	0.440
Pantothenic acid (mg)	3.082 ± 3.457	1.513 ± 0.757	0.121	0.440
Vitamin B6 (mg)	1.342 ± 1.742	0.565 ± 0.315	0.342	0.440
Folate (µg)	656.6 ± 1756.8	146.9 ± 57.23	0.178	0.440
Vitamin B12 (µg)	1.674 ± 1.331	1.321 ± 0.8019	0.827	1.000
Vitamin C (mg)	77.54 ± 55.70	38.37 ± 13.42	0.121	0.440
Vitamin D (µg)	0.2417 ± 0.2356	0.2060 ± 0.2813	0.368	0.440
Vitamin E (mg)	3.486 ± 4.169	1.698 ± 0.9425	0.544	1.000
Vitamin K (µg)	73.60 ± 51.83	45.08 ± 19.44	0.195	1.000
Calcium (mg)	422.8 ± 552.7	191.14 ± 73.89	0.099	0.440
Copper (µg)	1.819 ± 3.654	0.4256 ± 0.1861	0.162	0.440
Iodine (µg)	13.27 ± 11.87	7.499 ± 3.334	0.212	0.120
Iron (mg)	16.52 ± 30.10	6.551 ± 2.555	0.212	1.000
Magnesium (mg)	280.28 ± 511.77	97.31 ± 44.68	0.121	0.440
Phosphor (mg)	1061.5 ± 793.3	445.9 ± 194.9	0.394	1.000
Potassium (g)	2526.9 ± 3433.3	1035.5 ± 399.5	0.272	0.440
Sodium (g)	1528.8 ± 863.5	1063.3 ± 485.5	0.272	1.000
Zinc (mg)	13.13 ± 16.31	6.855 ± 2.372	0.610	0.704

## Data Availability

Data and materials will be provided by the corresponding author upon request.

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
