# Peer review of "The Nutritional Content of Rescued Food Conveyed by a Food Aid Organization"

_ijerph, 2021, doi:10.3390/ijerph182212212_

Round 1

Reviewer 1 Report

The problem, gap in the current literature and methodology were presented well.  The article was easy to read and follow, and for my purposes, appears to be complete in all respects (I am not a nutritionist, but expert in food insecurity and  food assistance programs).   I was surprise by the results, and think the findings are an important contribution to the understanding of the nutritional value of food rescue efforts.   It points a clearly different policy framework than other, more traditional forms of food assistance.  I see two major limitations: 1) the sample size is small and 2) the food rescue context in Portugal may not be generalizable to many other countries.  In terms of the sample size, it would almost always be better to have larger samples.  However, the implementation barriers in studying this issue in this way are large - I am impressed with the thoroughness achieved as it is.   In this regard, I would suggest to the editors and authors that this would serve as an example of an excellent pilot study on which to base funding requests for a larger research effort.   The authors could speak more to how this limitation is an opportunity to present the next round of research, and what would be done differently to improve going forward.   In terms of the generalizability, I am concerned that the sample was taken from only one food rescue organization, which may have relationships with the donor organizations that are not typical in other communities, even within Portugal.  More details on how/why this case study has value for the larger questions would help.   I am not bothered that it is a case study at all.  However, even a case study has to discuss why it speaks to the larger issue - why should readers in the UK, US, Greece, Columbia or Sweden pay attention?  To me, the key issues are that 1) the food, especially the prepared food, is likely to be culturally appropriate, and more likely eaten; 2) the barriers presented by lack of cooking time, tools, or knowledge are avoided to some degree; and 3) the surprising result that challenges prior work showing low nutrition in many traditional shelf-stable or bulk food distributed foods.   So even a limited case study from Portugal should make everyone sit up and take notice and change the direction of research.   I recommend publishing.

Author Response

We would like to thank the reviewer for the valuable and very positive comments and relevant suggestions. We have made changes according to the suggestions and hope to have satisfied all requests and suggestions.

Detailed responses to each comment and suggestion are provided in the attached file. The major changes in the revised version are in green in this response and in the revised manuscript.

We look forward to hearing from you.

Kind regards,

Anne Nogueira

Reviewer 2 Report

The subject of this study is relevant and pertinent, however the paper needs several improvements. 

Abstract

  • should mentionate the tool used to measure "perception of food insecurity"

Introduction

  • some sentences had double space between words (line 62 / 87) or references in superscript (line 92)

Materials and Methods

  • Line 131 - visited 23 times during 5 months, with what frequency per month? This could be clarified. The process of data collection during each visit is not clear.
  • Line 134 - 139 . This section should be improved. It is not clear if the data collection corresponds to total food distributed during this period, or if the families are the total families beneficiary of this program during this period. If was only a sample of subjects or food, how they are selected? Which criteria?
  • Line 168 - the researchers have not weigh every basket content before going to each family? it is not clear.
  • Line 174, it is important know if the families choosed for this studied collect food "1, 2 or 3 times per week" and if this is taken in consideration in results analysis.
  • Line 199 - "height"
  • Line 200-201 - for children, the BMI percentile curves should be considered in the classification of nutritional status. Please report how it was done and review the results in table 1.
  • Major concern: This study was previously approved from a Ethical Committe? The participants had subscrived an informed consent?

Results

  • Lines 232-233, this process should be clarified. Why don't calculate the nutritional needs for children if the nutritional requirements are different?
  • Table 2 - This results are to the 59 subjetcs? including children? why don't have calculated the nutritional needs per person / day, considering their age and reported heigh and weight? Several DRI are different to adults and children... it is not correct merge adults and children and say that they achieve xx % of daily need of specific nutrient.
  • Lines 259-261 - for children the BMI should be analysed in percentile...according to the percentile BMI curves per age.
  • Line 298 - references in superscript?
  • Line 385 - it will be nice if reported weight and heigh was considered also a limitation.

Author Response

We would like to thank the reviewer for the valuable and relevant suggestions. We have made changes according to the suggestions and hope to have satisfied all requests and suggestions.

Detailed responses to each comment and suggestion are provided in attached file. The major changes in the revised version are in blue in this response and in the revised manuscript.

We look forward to hearing from you.

Kind regards,

Anne Nogueira

Round 2

Reviewer 2 Report

I am satisfied with the improvements.

Author Response

We would like to thank the reviewer for the valuable and relevant suggestions. We have made changes according to the suggestions and hope to have satisfied all requests and suggestions.

Since Round 1, the English language and spelling was revised by professional services. A Total Methodology Flow Chart was added as an Appendix, to adequately describe the research design and methods. Please find attached the revised version.

We look forward to hearing from you.

Kind regards,

Anne Nogueira
